# Advances in High Throughput Proteomics Profiling in Establishing Potential Biomarkers for Gastrointestinal Cancer

**DOI:** 10.3390/cells11060973

**Published:** 2022-03-11

**Authors:** Md Zahirul Islam Khan, Shing Yau Tam, Helen Ka Wai Law

**Affiliations:** Department of Health Technology and Informatics, Faculty of Health and Social Sciences, The Hong Kong Polytechnic University, Hung Hom, Hong Kong, China; zahir.islamkhan@connect.polyu.hk (M.Z.I.K.); marco-shing-yau.tam@polyu.edu.hk (S.Y.T.)

**Keywords:** biomarkers, gastrointestinal cancer, mass spectrometry, proteomics, multi-omics

## Abstract

Gastrointestinal cancers (GICs) remain the most diagnosed cancers and accounted for the highest cancer-related death globally. The prognosis and treatment outcomes of many GICs are poor because most of the cases are diagnosed in advanced metastatic stages. This is primarily attributed to the deficiency of effective and reliable early diagnostic biomarkers. The existing biomarkers for GICs diagnosis exhibited inadequate specificity and sensitivity. To improve the early diagnosis of GICs, biomarkers with higher specificity and sensitivity are warranted. Proteomics study and its functional analysis focus on elucidating physiological and biological functions of unknown or annotated proteins and deciphering cellular mechanisms at molecular levels. In addition, quantitative analysis of translational proteomics is a promising approach in enhancing the early identification and proper management of GICs. In this review, we focus on the advances in mass spectrometry along with the quantitative and functional analysis of proteomics data that contributes to the establishment of biomarkers for GICs including, colorectal, gastric, hepatocellular, pancreatic, and esophageal cancer. We also discuss the future challenges in the validation of proteomics-based biomarkers for their translation into clinics.

## 1. Introduction

The management of cancer predominantly depends on early diagnosis, proper staging, and suitable selection of treatment modalities. Aspects of management such as risk assessment, screening using differential diagnosis, prognosis, tumor recurrence prediction, and treatment response evaluation can all be facilitated by analysis of cancer biomarkers [1,2,3]. Because of the potential roles of biomarkers in all cancer stages, they pass through several evaluation processes including analytical, clinical, and instrumental validations prior to incorporation into clinical settings [1,4,5]. According to the definition of the National Cancer Institute (NCI), cancer biomarkers are biological molecules derived from our body that are differentially expressed or affected during carcinogenesis compared with the normal state [1]. Therefore, DNA, mRNA, micro-RNAs (miRNAs), proteins, exosomes, enzymes, and metabolites are commonly used cancer biomarkers (Figure 1). The sources of these biomarkers are not limited to the tumor itself and the blood. Biomarker detection has been performed in other body fluids, neighboring tissues, and secondary metastasis sites [6]. According to the definition, diagnostic biomarkers can be used to detect pathogenesis at an early stage, whereas prognostic biomarkers allow us to predict disease outcomes. The term therapeutic biomarkers, however, refers to the proteins or other biological molecules that could be used for treatment [7]. Appropriate categorization of biomarkers is essential in drug design and for delivery to the target site [8]. The development of reliable, cost-effective, highly sensitive, and specific biomarkers is the primary objective of the field of cancer biomarkers [9]. Hopefully, the most effective treatments can be designed for each patient and the clinicians can effectively monitor the treatment progress and outcomes. 

The screening and identification of potential biomarkers depend on many factors such as cancer type, associated microenvironment, tumor metabolic nature, associated body responses, and development of metastasis. Cancer metastasis is a complex multi-step process consisting of proteolytic activity, local infiltration of tumor cells into the specific tissue, migration, proliferation, and extravasation [10,11]. There is evidence that invading tumor cells neovascularize to stimulate the local existing blood vessels by interacting with multiple protein complexes. The protein-protein interaction, therefore, governs the mobility and plasticity of the tumor cells and stabilizes the tumor microenvironment, allowing it to settle and colonize [11,12]. For metastatic cancer, it is very important to examine the role of each regulatory protein or biological molecule associated with tumor progression and suppression. For example, it was previously demonstrated that synthetic glucocorticoid is commonly used as co-medication in leukemia and lymphoma, where glucocorticoid receptor (GR) interacts with other receptors to promote apoptosis. However, the activity of glucocorticoids expresses differentially in breast cancer. Recent studies have shown that glucocorticoids reduce cell proliferation in an ER-positive model and GR activation dramatically reduces the chemotherapy-mediated cytotoxicity in triple-negative breast cancer [13,14,15]. 

Over the recent decades, omics technologies have emerged as promising tools for discovering and identifying molecular pathways by absolute quantification of differentially expressed molecules associated with various pathological states [16,17,18]. In particular, omics technologies revolutionized cancer research where they can effectively identify the region of somatic mutations, transcript number variations, and gene expression profiling using genomic technologies [19]. The advance in knowledge and technologies of next-generation sequencing (NGS), gene expression arrays, and high-throughput mass spectroscopy (MS) allow prompt identification of biomarkers for individual cancer types [1,2,6,20]. These advanced techniques are generating many big databases. However, unsupervised data mining and screening without clear intentions may lead to reporting of false-positive biomarkers [1,6]. Despite recent progress in the development and screening of cancer biomarkers, there are still many issues to be addressed. Specifically, gene expression profiles must be studied carefully to establish novel non-invasive, specific, and sensitive biomarkers for cancer diagnosis, prognosis, and management [21].

Proteins are dynamic regulatory components of cells that maintain cellular processes. Under normal conditions, proteins are distributed throughout the body in their native structures and maintain homeostasis by regulating cellular complex pathways. The normal functions of proteins or proteostasis may be dysregulated in response to stress, resulting in abnormal housekeeping activity in the cells [22]. Among all types of biomarkers, proteins can be detected at nanogram to microgram levels. The high sensitivity of detection made them excellent potential candidates for biomarkers of individual types of diseases [22,23]. Unlike proteomics analysis, other recent technologies including, genomic and transcriptomic analysis, cannot reflect the changes in the final cellular regulatory product of the central dogma. These recent technologies can be used to predict expression changes of proteins and their isoforms which may regulate the biological processes of cells. However, proteomics analysis remains the most acceptable technique to explore underlying cellular mechanisms in the disease state [24,25].

The theory of abnormal protein expression being associated with various pathogenesis is well accepted. Nevertheless, the term proteomics cancer biomarker was only introduced in the last decade after the advancement of technologies related to protein study [26,27]. Proteomics research mostly studies the structure and functions of proteins, functions during pathogenesis, and the nature of individual proteins in each cell type. Proteomics biomarkers are usually protein components derived from the body’s own system during normal or pathological conditions. They can be efficiently identified and detected using techniques such as MS [26]. Nowadays, although genomic data is readily available, the identification and discovery of biomarker candidates are limited due to the heterogeneity of patients and the development of mutations in different cancer stages [28]. Therefore, homogenizing the genomic and proteomic analysis to form proteogenomics can be a potential strategy for the identification of potential biomarker candidates for numerous cancer types [19]. In this review, we highlight the recent advances in MS technologies and their potential uses in biomarker discovery in various GICs. We aim to provide a comprehensive review covering the existing challenges, recent developments in identifying biomarker candidates, and the potential application of MS-based biomarkers for identifying highly specific and sensitive diagnostic biomarkers with their potential perspectives in clinical translation.

## 2. MS Workflow and Recent Advancements

In cancer research sample collection mostly depends on the type and stage of cancer. An experimental proteomics workflow of bottom-up MS is illustrated in Figure 2. In the first step, proteins are extracted from patient samples or experimental models such as cancer cell lines or in vivo animal xenografts. Subsequently, two mainstream approaches including top-down and shotgun/bottom-up are used to analyze the proteomes based on the scientific questions [19,29]. The top-down proteomics approach provides the highest molecular precision for analyzing the primary structures of proteins without enzymatic digestion. Usually, the top-down strategy consists of two-dimensional protein separation techniques which isoelectrically focus and visualize the proteins in polyacrylamide gel electrophoresis (PAGE), in a technique called two-dimensional differential gel-electrophoresis (2D-DIGE). [30]. On the other hand, the shotgun approach provides an indirect quantification of proteins where the isolated proteins are proteolytically digested in order to make a peptide complex mixture that is further fractionated and subjected to LC-MS/MS analysis [31].

To establish a highly specific and sensitive biomarker, the advancement of proteomics tools and their detectability from body fluids such as urine, blood, stools, or biopsy samples is essential. The use of body fluids is better accepted among patients because of the non-invasiveness and cost-effectiveness compared with other methods. However, despite recent technological developments in proteomics, challenges in proper sampling or sample preparation need to be overcome, especially the challenge of sample collection from body fluids that contain complex mixtures of proteins [32,33]. There are several advantages of using blood for biomarker studies such as high specificity, an easy and cost-effective sampling procedure, high stability during analysis and storage, and low invasiveness of collection [34]. Alternatively, usage of blood plasma or plasma proteins for developing or identifying potential biomarkers poses several difficulties such as the wide variety of protein containments, the low abundance of targeted proteins, and the vast patient variability [35,36]. In addition, the discovery of proteomics-based biomarkers is less reliable when using only a single tool. Therefore, to improve the sensitivity and specificity of proteomics-based biomarkers a combination of two-dimensional gel electrophoresis (2D-PAGE), matrix-assisted laser desorption ionization-time of flight mass spectrometry (MALDI-TOF), surface-enhanced laser desorption ionization-time of flight mass spectrometry (SELDI-TOF), quadrupole time of flight mass spectrometry (Q-TOF), and isobaric tag for relative and absolute quantification (iTRAQ) or isotope-coded affinity tag (ICAT) and stable isotope labeling with amino acids in cell culture (SILAC) MS techniques are commonly used to establish error-free biomarkers [34,35,36,37,38].

Since the turn of the century, the continuous improvement in MS technologies was noticeable to all because of the rapid advancements in resolution, accuracy, data processing, and quantitative analysis [39,40]. The significant improvement in MS resolution provides precise molecular structural information. Therefore, we can accurately determine the biomolecular structure of specific proteins and their nature in a large-scale manner. Moreover, structural information and ion distribution play important roles in biomolecular dynamics which can also determine the sequential alteration of proteome expression in diverse molecular pathways. The quantitative analysis of these changes can be used as a direct target of drug discovery or act as diagnostic or therapeutic biomarkers clinically [39,40]. Apart from this, the recent extension of ion mobility spectrometry and high field asymmetric ion mobility spectrometry coupled with Orbitrap Eclipse is shown to offer increased sensitivity [41,42]. Indeed, single cell proteomics and targeted proteomics are recent advancements in the MS field which enable quantitative and targeted protein profiling with higher sensitivity, accuracy, and reproducibility from single cells or targeted cells [43,44].

It has previously been discussed that cancer biomarkers can be anything secreted biologically from the body in a regular or irregular manner (such as enzymes, hormones, receptors) or genetic alterations which significantly take part in carcinogenesis or work as oncogenes [27]. So far, thousands of proteins have been reported as potential cancer biomarkers, but only a negligible number of these were approved by US Food and Drug Administration (FDA) for clinical practice. The most commonly used MS-identified biomarkers are OVA1, pre-albumin, apolipoprotein A1, and transferrin. OVA1 was the first FDA-approved diagnostic biomarker for ovarian cancer (OVC) [45]. Some circulatory proteins such as CA15-3 for breast cancer, PSA for prostate cancer, and CA-125 for OVC have already been approved by FDA and are commercially available for diagnostic purposes [46,47,48]. Here, we summarize the proteomics-based cancer biomarkers proposed by recent studies for common GICs according to their global incidence.

## 3. Proteomics-Based Biomarkers for GICs

### 3.1. Colorectal Cancer (CRC)

CRC is common cancer with an annual incidence of nearly two million cases worldwide [49,50]. However, the majority of cases are from developed and western countries (e.g., USA, Europe, Australia). Disappointingly, the numbers for lower-economic, lower-middle economic, middle-economic, and economically developing countries have been rapidly increasing in the past decade [20,51]. The progression from sporadic CRC stage to advanced metastatic stage can take more than a decade, so early detection of benign polyps by regular screening can effectively improve the prognosis [52]. The prognosis of CRC is influenced by the cancer stage. The curability of early stages of CRC can exceed 90.0% following surgical removal of polyps. However, approximately 25.0% of the diagnosed patients have already developed distant metastases and their corresponding 5-year survival rate is less than 14.0% [53]. The current gold standard for CRC screening involves colonoscopy and fecal occult blood tests (FOBT). Colonoscopy is expensive, yet it has higher sensitivity in detecting polyps or adenoma formed in the rectum wall and facilitates complete resection of polyps during a single session [54]. Patient compliance for colonoscopy is currently reducing due to its high cost, invasiveness, and associated secondary complications such as perforation and bleeding [54]. Recently, the non-invasive FOBT screening approach has been extensively used due to its simplicity and lower cost. The replacement of colonoscopy by FOBT is still not perfect, as it has relatively high false positive and false negative rates and unsatisfactory selectivity and sensitivity [55].

Several LC-MS studies suggested various biomarkers for CRC (Table 1). Quesada-Calvo et al. proposed and validated KNG1, OLFM4, and Sec24C as diagnostic biomarkers among 561 differentially expressed proteins [56]. Another study revealed that ACTBL2, Aldose A, Annexin A2, cyclophilin A, and DPEP1 may also serve as biomarkers and new therapeutic targets for the management of CRC [57,58,59]. On the other hand, circulating proteins are widely accepted biomarkers sources for many pathological conditions including CRC due to higher abundances [60]. Several studies proposed that MRC1 and S100A9 are upregulated in the serum of CRC patients compared to healthy controls with western blot (WB) and ELISA for verification uses [61]. Similarly, Ivancic et al. revealed that CRC blood containing LRG1, EGFR, ITIH4, HPX, and SOD3 showed 89% specificity and 70% sensitivity in detecting CRC and that the proteins GC, CD44, CRP, and ITIH3 can potentially distinguish CRC according to their stages [62]. Moreover, Bhardwaj and co-workers revealed that five protein signatures, including AREG, MASP1, OPN, PON3, and TR, exhibited better diagnostic performance compared to the traditional FDA-approved blood-based CRC biomarkers [63]. Another study found that the downregulation of MST1 mRNA expression in plasma is associated with a poor prognosis of CRC. Importantly, MST1 can distinguish the patients by stage and shows adequate diagnostic efficacy in detecting Stage I CRC from the healthy control group [64]. Similarly, a recent study by Chantaraamporn and colleagues revealed that the plasma content of complement C9 and fibronectin may provide crucial information on CRC development [60].

Furthermore, Peltier et al., revealed that serum content of SERPINA1, SERPINA3, and SERPINC1 are potential biomarker candidates for CRC early detection. In this study, the sensitivity and specificity for SERPINA1 and SERPINC1 reached 95% [65]. More recently, Thorsen and colleagues performed a 2D gel-based MS study on 128 tumors with their adjacent normal biopsy samples. Their results revealed that TPM3 expressions validated using patient plasma samples could be potential diagnostic biomarkers for CRC [66]. Ludvigsen et al., primarily performed 2D-PAGE LC-MS/MS analysis on CRC cancerous cell line HCT-116 and CRC normal cell NCM460 to establish potential diagnostic biomarkers. The MS analysis and WB validation of S100A4, S100A6, RBP, SET, and HSP90B1 proteins from cell lines and patient samples proposed their use in clinical settings for the early diagnosis of CRC [67]. Another group performed an LC-MS/MS-based glycomics study and demonstrated that the N-glycome landscape of CRC cells exhibits aggressive metastatic nature in CRC pathogenesis by regulating the EGFR regulatory pathway [68]. Another recent study revealed that initial CRC development can be distinguished by MS-based post-transcriptional modification analysis of CRC plasma samples [69]. Early-stage CRC development is highly associated with dysregulation of cytokines and extracellular matrix proteins and reduction of extracellular matrix stability. The study revealed that APOE, APOC1, and APOB potentially impact DNA repair through mTOR and PI3K pathways and impact post-transcriptional modification. Therefore, these proteins may be used as potential biomarkers for the early diagnosis of CRC [69].

**Table 1 cells-11-00973-t001:** Representative Proteomics-based Biomarkers for CRC.

Sample Sources	Labeling and MS Methods	Proposed Proteomics-Based Biomarkers (Specificity:Sensitivity %:%)	References
Cell lines and tissues	2D PAGE, LC-MS/MS	S100A4, S100A6, RBP, SET, and HSP90B1	[67]
Cell lines and Tissues	LC-MS/MS-based N-glycomics	N-glycomes	[68]
Blood/serum	LC/multiple reaction monitoring (MRM)-MS	AREG (80:71), MASP1 (80:71), OPN (80:71), PON3 (80:71), and TFR1 (80:71)	[63]
Serum	LC-MS/MS	EGFR (70:89), HPX (70:89), ITIH4 (70:89), LRG1 (70:89), and SOD3 (70:89)	[62]
Serum	MALDI-TOF MS	MST1 (93.8:82.4)	[64]
Serum	2D LC-MS/MS	MRC1 and S100A9	[61]
Serum	iTRAQ/MALDI-TOF MS	SERPINA 1 (95:95), SERPINA 3 (55:95), and SERPINC1 (95:95)	[65]
Tissues	2D-DIGE + MALDI-MS	ACTBL2	[58]
Tissues	Nano-spray LC-MS/MS	DPEP1	[59]
Tissues and Plasma	LC-MS/MS	Aldolase A, Annexin A2, A1AG1, Complement component-9 (92:63), Cyclophilin A, Fibronectin (92:69), KNG1, OLFM4, and Sec24C,	[56,57,60]
Tissue and Plasma	MALDI-TOF	TPM3	[66]
Plasma	Liquid chromatography-mass spectrometry (HPLC-MS/MS)	APOE, APOC1, and APOB	[69]

### 3.2. Gastric Cancer (GC)

GC ranks among the top three cancers for cancer-related deaths worldwide with approximately 0.8 million deaths in 2018 [70]. The incidence rate of GC has decreased slightly over the last decade, but the 5-year survival worsened, now being less than 10% for those with advanced stages of GC. Apart from this, early diagnosis of GC increases the 5-year survival to 50% in developed countries [71]. Traditionally, carbohydrate antigen (CA) series CA19-9, CA24-2, CA50, CA72-4, and CA125, and CEA are the most frequently used tumor markers but none of them are being used clinically due to poor specificities and sensitivities [72]. Moreover, the complex heterogeneous nature, genetic mutations, translations, and post-translation alterations of GC are the key factors causing poor prognosis [72]. With the use of advanced technologies in molecular biology, GC is subdivided into four subclasses to improve the early diagnosis and prognosis of the diseases. These include tumors with Epstein-Barr positive virus, tumors with microsatellite instability, tumors with genomic stability, and tumors with chromosomal instability [73]. The new classification of GC provided an opportunity to investigate novel molecular therapeutic regimens combined with or without immune suppressors to improve outcomes through early diagnosis [72,74].

The proteomics approach is exceptionally promising as it deals with the functional genomic contents which participate in translational processes of initiation, progression, and metastasis of GC (Table 2) [75]. A study revealed that three protein peaks at 2873, 6121, and 7778 *m*/*z* from GC serum samples can distinguish the stages of GC [76]. Another study revealed that gastric juice contents of S100A9, GIF, and AAT proteins can facilitate diagnosis and predict the progression of GC [77]. Likewise, Liu and colleagues demonstrated that free amino acid profiling of patients’ gastric juice could be a potential approach for early detection of GC, achieving a high area under the curve (AUC) value [78]. Several research studies revealed that analysis of blood from GC patients is another potential source of biomarkers [79]. Wu and colleagues performed MS analysis and identified peak 5910 (*m*/*z*) as a potential candidate serological biomarker for the early diagnosis of GC [80]. Similarly, Cheng and colleagues found that a higher plasma level of SHBG can potentially distinguish GC patients from healthy ones [81]. Another recent publication screened 11 differentially expressed proteins from GC plasma samples as early diagnostic biomarkers for GC [82]. Alternatively, MS-based analysis of GC cancerous and normal serum has shown that peaks at 2953 and 1945 Da have higher specificity and sensitivity than traditional CEA and CA19-9 biomarkers in GC diagnosis [83]. Another recent study proposed that DEK proteins might be used as novel biomarkers for GC [84] finding that the plasma content of DEK also reached superior sensitivity compared to the traditional biomarkers CEA, CA19.9.

The protein analysis of GC tumors is another source of potential biomarkers. It has been revealed that the abnormal expression of several proteins is associated with GC initiation, progression, and metastasis. For example, LPCAT1, ANXA1, NNMT, Fibulin-5, UQCRC1, MTA2, and HDAC1 are differentially expressed in GC tumors compared with the adjacent normal tissues. These proteins may be used to establish diagnostic and prognostic biomarkers for GC [85,86,87]. Loei and co-workers demonstrated that expression of GRN occurs in GC tumors but not in normal tissues. This may also be used as an indicator for early-stage GC [88]. Jiang and colleagues performed quantitative proteomic analysis and found that co-expression of FABP1 and FASN can significantly distinguish GC normal samples from GC tumors [89]. Similarly, another recent study reported that GLS1 and GGCT were significantly upregulated in GC tissues compared to normal tissues. The overexpression is correlated with patients’ histological grade, TNM stage, and lymph node metastasis of GC [90]. An experimental demonstration found ITIH3 abundantly expressed in mice xenograft plasma samples, and subsequent validation of ITIH3 expression in clinical samples supported its clinical use for GC detection [91]. Another recent study reported that exosomal tripartite motif-containing protein 3 (TRIM3) is downregulated in GC tumors compared to adjacent normal samples. The overexpression of TRIM3 can suppress GC cell growth and suppress the development of metastasis both in vitro and in vivo [92]. Their findings suggested that TRIM3 could serve as a potential diagnostic biomarker for GC as well as suggesting that efficient delivery of TRIM3 can provide new targets for GC.

**Table 2 cells-11-00973-t002:** Representative Proteomics-based Biomarkers for GC.

Sample Sources	Labeling and MS Methods	Proposed Proteomics-Based Biomarkers (Specificity:Sensitivity %:%)	References
GC cell lines	MALDI-TOF MS	CIP2A, PIK3CB	[93]
Plasma and cell lines	iTRAQ	DEK (79:70.4)	[84]
Mice and cell lines	iTRAQ/ LC-MS/MS	ITIH3 (66:96)	[91]
Gastric juice, Plasma, Serum	LC-MS/MS	ANK1 (86.7:46.7), FOLR2 (80:60), Gastric juice free amino acid (89.2:85.1), GRN, LILRA2 (93.3:60), MGP (80:73.3), NBL1 (53.3:80), OAF (100:46.7), PCSK9 (80:60), PSTPIP2 (53.3:73), RPS27A (66.7:86.7), SHBG, SOD1 (46.7:93.3), and TRIM3	[78,81,82,88,92]
Gastric fluids, Serum, tissues	MALDI-TOF MS, 2D-DIGE MALDI-TOF MS, SELDI-TOF MS	AAT, CIP2A, GIF, LPCAT1, PIK3CB, S100A9, peaks at Da of 2863 (75:75), 2953 (85:85), 1945 (90:90), and 2082 (75:75), and peaks at m/z 5910 (91.3:86.3), 5342 (80.6:80.3), 6439 (70.3:73.3), 2873 (91.57:93.49), 3163 (91.57:93.49), 4526 (91.57:93.49), 5762 (91.57:93.49), 6121 (91.57:93.49), and 7778 (91.57:93.49)	[76,77,80,83,85,93]
Tissue and plasma, mouse plasma	iTRAQ, 2D-DIGE LS/MS, LC-ESI-MS/MS	ANXA1, FABP1 (77.1:61.4), FASN (77.1:61.4), Fibulin-5, GGCT (60.7:63.1), GLS1 (81:75.6), HDAC1 (63.2:61.8), MTA2 (55.3:57.9), NNMT, and UQCRC1	[84,86,87,89,90,91]

### 3.3. Hepatocellular Carcinoma (HCC)

Liver cancer is the sixth most frequently diagnosed cancer globally, of which 90% were HCC [94]. Despite significant improvements in diagnosis and management, the 5-year survival rate of HCC is still less than 18%. The poor prognosis of HCC is largely due to late diagnosis, intrahepatic metastasis, and the development of resistance to conventional therapies [95]. The diagnosis of HCC primarily relies on imaging or histopathological characteristics of patient samples. In recent years, the advances in omics-based technologies have shown potential in identifying early diagnostic biomarkers for HCC (Table 3). For example, Guo et al. identified 93 differentially expressed proteins from a patient cohort using iTRAQ, CD14 expression was validated with a specificity and sensitivity greater than 80% to establish an early diagnostic biomarker for HCC [96]. Kim and colleagues revealed that fucosylated peptide AFP can be a serum-based diagnostic biomarker for patients with HCC [97]. In addition, Ding and co-workers proposed salivary-based non-invasive biomarkers for HCC [98]. They identified 133 differentially expressed biomarkers from HCC patients and adjacent healthy controls. Finally, they sieved and verified SOD2 expressions by ELISA from a dataset and concluded that it can be a potential marker for HCC early detection. Likewise, another study demonstrated that AFP and ORM1 abundancy in urine samples can be a potential non-invasive diagnostic biomarker for HCC [99]. They ensured a higher predictive value and sensitivity at satisfactory levels of about 80%.

Alternatively, a recent trend in MS involves the identification of several peptides from serum samples as potential diagnostic biomarkers for HCC in clinical settings. For example, Heo et al. achieved specificity and sensitivity of around 80% for early diagnostic HCC marker EIF3A peptides from serum samples, and ELISA and western blotting validation confirmed the use of selected peptides clinically [100]. Similar to the previous study performed by Zhan’s group [99], Lee’s group also validated the presence of AFP in a large number of HCC patients’ serum samples and verified AFP expression by using RT-qPCR and western blotting techniques [101]. Furthermore, another study revealed that AFP along with heptoglobin specific N-glycopeptides may achieve a sensitivity of greater than 80% as diagnostic biomarkers for nonalcoholic steatohepatitis HCC patients [102]. Wu and colleagues revealed that triple quadrupole MS (QqQ MS)-based identification of miR-224 in HCC serum samples may also be a potential diagnostic and prognostic marker in clinical practice [103].

**Table 3 cells-11-00973-t003:** Representative Proteomics-based Biomarkers for HCC.

Sample Sources	Labeling and MS Methods	Proposed Proteomics-Based Biomarkers (Specificity:Sensitivity %:%)	References
Cells and serum	LC/ESI-MS/MS	BMP1 (71.4:90), FAP (90:76.2), EIF3A (83.5:79.4), and TRIM22 (85.7:90)	[100,101]
Serum	Q-TOF, TQMS	AFP (85.7:53.9) and miR-224	[97,103]
Serum, Saliva, Urine	iTRAQ	AFP (94.4:31.6), CD14 (50:94.7), SOD2, u-AFP (95.4:62.5), and u-ORM1	[96,98,99]
Serum	Electron-transfer/higher-energy collision dissociation (EThcD-MS/MS)	N-glycopeptides: N184_A3G3F1S3 (67:81), N241_A2G2F1S2, N241_A3G3F1S3 (73:81), N241_A4G4F1S3, and N241_A4G4F1S4	[102]

### 3.4. Pancreatic Cancer (PNC)

PNC is one of the most uniformly fatal cancers, for which approximately 85% of cases are classified as its major subtype, pancreatic ductal adenocarcinoma (PDC) [104]. In 2018, there were 458,918 newly diagnosed PNC cases with 432,242 deaths which ranked seventh for cancer-related deaths globally [70]. PNC has a 5-year survival rate of approximately 9%, which is the one of poorest prognoses among all types of cancer [70]. The most common reasons behind the reduced PNC survival are early systemic spread, difficulties in screening, and treatment failure [105]. Surgical removal is the first curative treatment modality, but only 15–20% of patients can receive up-front radical surgery, and the recurrence rate 2 years post-operation is more than 60% [106]. To overcome these difficulties, high-performance screening tests with adequate specificity and sensitivity greater than 95% are required. So far, the only FDA-approved biomarker for PNC is carbohydrate antigen (CA) 19-9, however, it is not considered as a PNC screening tool due to unsatisfactory specificity and sensitivity. The upregulation of CA 19-9 is not only associated with PNC but also with other pancreaticobiliary complications, such as severe pancreatitis or obstruction in the biliary. Moreover, 10% of the population are unable to express CA 19-9 due to reduced Lewis glycosyltransferase in their body [107]. However, CA 19-9, along with computed tomography and magnetic resonance imaging, is often used in PNC diagnostic settings but CA 19-9 alone is not recommended for diagnostic purposes [108].

To improve the specificity and sensitivity of PNC diagnostic biomarkers, researchers have focused on advanced proteomics approaches. The recent potential PNC diagnostic biomarker candidates have been listed in Table 4. Briefly, Xie et al. performed multiple array validation to achieve satisfactory diagnostic values for ApoA-I&II in NPC disease controls and healthy volunteer samples. Their demonstration revealed that ApoA-I&II can be used in clinical settings, having validated the findings using a large cohort of patients [109]. Similarly, Wu and colleagues identified PROZ and TNFRSF6B as novel serum-based diagnostic biomarkers for PNC which can differentiate the cancerous and non-cancerous patients according to their stage [110]. A promising glycoprotein panel was used to establish an early diagnostic biomarker where AACT, THBS1, and HPT AUC values reached satisfactory levels along with that for traditional biomarker CA 19-9 [111]. Likewise, Deutsch and colleagues revealed that CypB, a 21-kDa protein present in the saliva of PNC patients is a potential biomarker candidate with more than 90% AUC [112]. Takenami et al. revealed that five proteins including KRT17, ANXA10, TMEM109, PTMS, and ATP1B1 have promising diagnostic values which can distinguish pancreatic head cancer patients from distal cholangiocarcinoma [113]. More recently, it has been shown that exosomal ZIP4 can significantly promote the development and progression of PNC in vitro and in vivo. In addition, the AUC of ZIP4 proved that it may be used as a novel diagnostic biomarker for PNC [114].

### 3.5. Esophageal Cancer

Despite the recent advancements in esophageal cancer diagnosis and treatment, early diagnosis is still a key factor in the effective management of esophageal cancer (Table 5) [115]. To address the need for early diagnosis, a study revealed that the serological content of three differentially expressed proteins FLNA, TUBB, and UQCRC1 may serve as novel post-genomics putative biomarkers for early detection of esophageal squamous cell carcinoma (ESCC) [115]. Shah et al. identified 8 glycoproteins from esophageal adenocarcinoma patients’ serum samples with 94% area under curve (AUC) value, from which two candidates were verified for development as early diagnostic biomarkers using lectin magnetic bead array-based immunoblot in a large-scale patient cohort [116]. The application of high-throughput proteomics in ESCC serum samples and subsequent analysis by genetic algorithm models identified AHSG, FGA, and TSP1 circulating peptides as potential biomarkers for early detection of ESCC with sensitivity and specificity of more than 95% in a cohort of 477 patients [117]. The screening of patient plasma samples can also be used to establish early detection biomarkers in ESCC. Zhao and colleagues used ESCC plasma samples and performed a combined method DIGE with MALDI-TOF/TOF which identified AHSG and LRG circulating peptides that could be used to provide a framework for early ESCC screening [118]. In addition, the combination of iTRAQ and 2D-LC-MS MS identified ECM1 and LUM as potential plasma biomarkers for ESCC patients, followed by WB to validate the findings in vitro [119]. Subsequently, another study identified a significant decrease of PA28β from ESCC tissues, and the knock-in of PA28β was found to reduce tumor growth and proliferation in vitro. This might act as a useful biomarker for ESCC [120].

## 4. Conclusions and Future Perspectives

It is universally accepted that early diagnosis of cancers would not only improve patient prognosis but also facilitate a better understanding of the disease, reduce morbidity, and improve patient quality of life. The advances in proteomics techniques and quantitative analysis of MS data over the past decade have enabled the identification of each protein from a complex mixture of proteomes. These technological and analytical advances have contributed to the identification of cancer-specific biomarkers with higher specificity and sensitivity in GICs. The use of protein-based biomarkers for the diagnosis of GICs could be of importance considering the accessibility of proteins from clinical samples, such as blood, feces, gastric juice, urine, saliva, and sputum. However, the sensitivities and specificities of traditional biomarkers are not high in clinical practice. It is therefore suggested to use multiple protein panels rather than a single biomarker for diagnosing cancer. The many shortlisted candidates can be evaluated for potential clinical use through validation in a large sample cohort and an appropriate combination of biomarkers.

With the advent of the latest sequencing technologies, the field of cancer biology has become increasingly dependent on “multi-omics” data. The term multi-omics is a recent approach of data sets where different groups of omics data such as genome, epigenome, proteome, transcriptome, metabolome, and microbiome are combined during analysis [121,122]. The quantitative analysis of multi-omics data and clinical features can provide information about changes in molecular levels and allow a better understanding of complex biological pathways systematically and holistically [122,123]. The application of an integrated multi-omics approach can reveal the actual flow of information from one omics level to another. As a result, it will help us in bridging and fulfilling the gap between genotypic and phenotypic levels. Eventually, it will enhance the accuracy of diagnosis, prognosis, treatment, and prevention of cancer. Due to the large volume of data available, it is necessary to employ big data analysis and use the multi-omics approach to link all the information together. For example, patient demographic, genomic, proteomic, radiomic, and microbiota data can be integrated to facilitate the design of personalized treatment and prediction of clinical outcomes.

Another new area of multi-omics is radiomics which analyses the medical imaging data associated with cancer development and outcomes. Imaging biomarkers are one of the most commonly used approaches in clinical settings due to their non-invasiveness, availability, and cost-effectiveness. The response evaluation criteria in solid tumors (RECIST) is considered a gold standard for evaluating treatment outcomes of solid tumors. Imaging biomarkers enable monitoring of tumor changes over time, can identify the spatial heterogeneity of tumors, and can assess multiple different lesions of the tumors. However, multiple challenges such as image standardization for multi-center clinical trials, image acquisition, image processing, and image analysis must be considered before establishing image-based biomarkers [124]. Moreover, a public database could be compiled by submitting proteomics or multi-omics analysis data from various cancer types to facilitate the rapid diagnosis and prognosis of cancer. Importantly, we may use artificial intelligence (AI) algorithms to resolve the complex intractable problems in cancer [125].

Circulatory tumor cells (CTCs) are another recent approach in cancer biomarker discovery. Studies showed that CTCs have great potential in revealing metastasis processes including extravasation of CTCs from primary sites, the establishment of tumor microenvironment by communicating other cells or proteins, and intravasation of tumors to the distant sites [126]. Therefore, it is believed that CTCs can greatly contribute to the discovery of metastatic cancer biomarkers and can potentially identify the targets for anti-metastatic therapies. So far, various technologies have emerged to identify the CTCs, however, further, improvement is mandatory to increase the specificity and sensitivity of CTCs as a biomarker or therapeutic targets.

Recently, immunotherapy has become one of the preferred treatments for cancer. Chemotherapy and other anti-cancer agents directly or indirectly target the cancer cells to promote cancer cell death whereas immunotherapy activates the host immune system in order to promote targeted killing of cancer cells and eliminate them from the body. This controls the tumor microenvironment and promotes the activity of anti-cancer therapeutics [127,128,129]. Protein profiling can be done for those patients receiving immunotherapy to predict their survival, and ultimately, prognostic biomarkers and proteins responsible for immunotherapy resistance may be identified [130,131].

Based on the previous studies on biomarker discovery for GICs, numerous obstacles need to be overcome before translation into clinical practice. Despite advances in MS techniques and analysis, there is still an absence of strategies and selection panels for the evaluation of candidate biomarker specificities and sensitivities. Another predominant factor in discovering non-invasive diagnostic biomarkers is the dynamic complexity of the proteome contents in blood, serum, urine, gastric juice, and feces. In addition, there is difficulty in the validation of biomarker candidates in large cohorts of patients because the patient data usually comes from different laboratories. The data may be further validated through performing meta-analysis to establish potential diagnostic biomarkers. The heterogeneity of patients and their sporadic cancers is another major obstacle to overcome. Understanding the biological and molecular heterogeneity of disease states by performing advanced MS at single-cell resolutions may tackle this problem.

## Figures and Tables

**Figure 1 cells-11-00973-f001:**
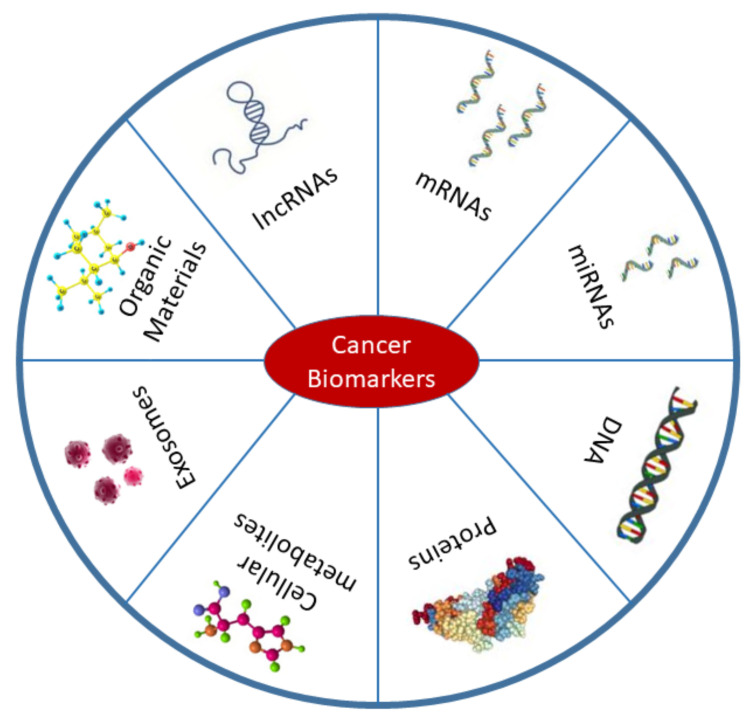
Types of cancer biomarkers. Biomarkers are mostly found in body fluids including blood, urine, saliva, and also in cancer tissues. Cancer biomarkers belong to a variety of biological elements such as DNA, mRNAs, proteins, long non-coding RNAs (lncRNAs), miRNAs, exosomes, cellular metabolites, and organic materials. This figure was based on a published article by Wu et al. (2015) [9].

**Figure 2 cells-11-00973-f002:**
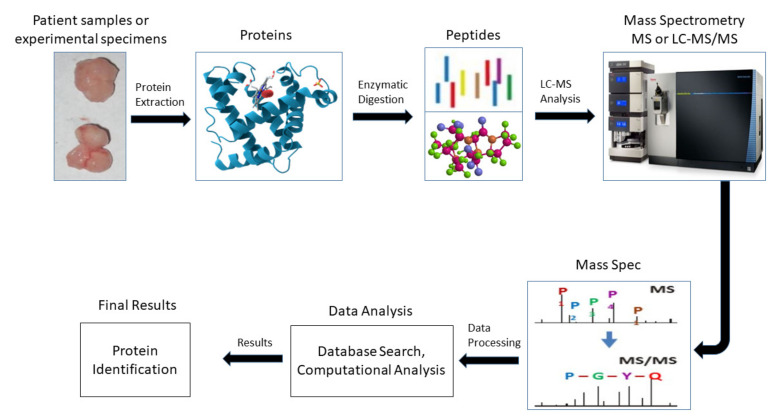
A typical workflow of bottom-up mass spectrometry. Proteins are collected from patients or experiments, and then chemically or enzymatically digested for conversion into the mixture of peptides. The peptides are then ready for mass spectrometry instrumentation and separated into a spectrum according to the mass value. Finally, the proteins are analyzed by using database search or computation analysis.

**Table 4 cells-11-00973-t004:** Representative Proteomics-based Biomarkers for PNC.

Sample Sources	Labeling and MS Methods	Proposed Proteomics-Based Biomarkers (Specificity:Sensitivity %:%)	References
Cell lines and mice tumors	Nano LC-MS/MS	Exosomal ZIP4	[114]
Serum	SELDI-TOF MS	ApoA-I and ApoA-II	[109]
Serum	iTRAQ	PROZ (95:79), TNFRSF6B (82.5:90.2), and CA-19-9 (87.5:71.4)	[110]
Serum, tissues, and saliva	LC-MS/MS	AACT (80:75.6), THBS1 (65.7:77.5), HPT (56.7:85.5), CA 19-9 (77.1:82.5), CypB, KRT17 (71.6:76.4), ANXA10 (51.3:81.9), TMEM109 (63.5:66.7), PTMS (72.2:60.8), and ATP1B1 (58.3:60.8)	[111,112,113]

**Table 5 cells-11-00973-t005:** Representative Proteomics-based Biomarkers for Esophageal Cancer.

Sample Sources	Labeling and MS Methods	Proposed Proteomics-Based Biomarkers (Specificity:Sensitivity %:%)	References
Plasma, cell lines, and tissues	MALDI-TOF	AHSG, LRG, PA28β	[118,120]
Serum	MALDI-TOF	FLNA (96.88:95.83), TSP1 (95.92:97), TUBB (96.88:95.83), and UQCRC1 (96.88:95.83)	[115,117]
Serum	Q-TOF	26 lectin–protein candidates	[116]
Plasma	iTRAQ	ECM1	[119]

## Data Availability

Not Applicable.

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
