# Peer review of "Advances in High Throughput Proteomics Profiling in Establishing Potential Biomarkers for Gastrointestinal Cancer"

_cells, 2022, doi:10.3390/cells11060973_

Round 1

Reviewer 1 Report

The present manuscript accessed the biomarkers for GICs including, colorectal, gastric, hepatocellular, pancreatic, and esophageal cancer. However, the gastrointestinal endoscopy has been widely carried out in clinics. The methods for diagnosis of colorectal and gastric cancer have been improved. Therefore, I have some major revisions.

  1. Abstract: Line 10-11, the treatment outcomes of GICs are not limited. The description is not accurate.
  2. There is little new in Figure 1 and Figure 2. They could be shown in supplementary figures.
  3. Recent advancements in MS were not detailed.
  4. The biomarkers could be divided into cells, animals and patients.
  5. Tables should better to show sensitivity and specificity.

Author Response

Point to point reply to the reviewer-1

Comment-1: Abstract: Line 10-11, the treatment outcomes of GICs are not limited. The description is not accurate.

Reply: We apologise for being unclear and have corrected the statement in the revised manuscript as “The prognosis and treatment outcomes of many GICs are poor because most of the cases are diagnosed in advanced metastatic stages.

Comment-2: There is little new in Figure 1 and Figure 2. They could be shown in supplementary figures.

Reply: Both Figure 1 and Figure 2 served as summaries of existing knowledge, and we prefer to retain them in the main text.

Comment-3: Recent advancements in MS were not detailed.

Reply: We appreciate Reviewer #1’s suggestion. We have added “Recent advancements in MS” in Paragraph 1&3 of Section 2 in the revised manuscript.

Comment-4: The biomarkers could be divided into cells, animals and patients.

Reply: We have followed the suggestions and revised all the tables accordingly.

Comment-5: Tables should better to show sensitivity and specificity.

 Reply: We really thank and appreciate Reviewer #1’s suggestions. Considering the suggestions, we reviewed all the biomarker candidates from Table 1 to 5 and the available specificity & sensitivity data are included in the revised tables accordingly.  

Reviewer 2 Report

The authors focused on on the advances in mass 18 spectrometry along with the quantitative and functional analysis of proteomics data that contributes to the establishment of biomarkers for GICs.

Proteomics techniques have been utilized to understand how the signaling pathways in tumor cells are altered. In future, this would help us to target specific pathways and treat cancers effectively. The authors need to address such pathways.

Multi-Omics approaches using patient samples are very important in translational research. The authors need to address such issues.

The authors need to describe pro-metastatic transcription factor.

The activity of the glucocorticoid receptor (GR) at metastatic regions which could affect the survival rate, need to addressed.

The proteomics approach can be employed to know the characteristics of any drug resistance and discover drug targets. These facts were missed out in the review.

More details of proteomics approach to immunotherapy are needed.

Data obtained from proteomics analyses of various types of cancer could be used to create databases. The authors need to discuss such issues.

The authors could add facts on artificial intelligence (AI). AI can be implemented to create algorithms that could increase their performance.

How can application of AI to omics help in target profiling and integration in cancer? These are importance facts which need to be discussed. 

Histology-based prediction models could be useful.

Author Response

Point to point reply to the reviewer 2

Comment-1: The authors focused on the advances in mass spectrometry along with the quantitative and functional analysis of proteomics data that contributes to the establishment of biomarkers for GICs.

Reply: N/A

Comment-2: Proteomics techniques have been utilized to understand how the signaling pathways in tumor cells are altered. In future, this would help us to target specific pathways and treat cancers effectively. The authors need to address such pathways.

Reply: We thank Reviewer #2 for the suggestion. We have revised the manuscript and addressed this issue in Section 2, Paragraph 3.

Comment-3: Multi-Omics approaches using patient samples are very important in translational research. The authors need to address such issues.

Reply: We appreciate Reviewer #2’s comments. We have addressed this issue in Section 4, Paragraph 2&3.

Comment-4: The authors need to describe pro-metastatic transcription factor.

Reply: This issue has been added to the manuscript Section 1, Paragraph 2.

Comment-5: The activity of the glucocorticoid receptor (GR) at metastatic regions which could affect the survival rate, need to addressed.

Reply: We have now discussed the diversified role of glucocorticoid receptor in various tumor type in Section 1, Paragraph 2.

Comment-6: The proteomics approach can be employed to know the characteristics of any drug resistance and discover drug targets. These facts were missed out in the review.

Reply: We appreciate Reviewer #2’s comments and we have addressed the role of proteomics approach in drug resistance and discovery of therapeutic targets in Section 2, Paragraph 3.

Comment-7: More details of proteomics approach to immunotherapy are needed.

Reply: We have enriched the discussion by addressing the role proteomics approach in immunotherapy in Section 4, Paragraph 5 of the revised manuscript.

Comment-8: Data obtained from proteomics analyses of various types of cancer could be used to create databases. The authors need to discuss such issues.

Reply: This issue has been added to Section 4, Paragraph 3.

Comment-9: The authors could add facts on artificial intelligence (AI). AI can be implemented to create algorithms that could increase their performance.

Reply: The implementation of AI together with proteomics approach have been discussed in Section 4, Paragraph 3.

Comment-10: How can application of AI to omics help in target profiling and integration in cancer? These are importance facts which need to be discussed. 

Reply: This issue has been included in Section 4, Paragraph 3.

Comment-11: Histology-based prediction models could be useful.

Reply: This issue has also been addressed in Section 4, Paragraph 3.

Reviewer 3 Report

Very well organized and comprehensive review of topic

My one major critique is the omission of a section on esophageal/GEJ adenocarcinoma - this felt skipped over as there was a section on Gastric Cancer and a section on esophageal squamous cell cancer.  Perhaps there is nothing to review for this type of cancer, but its omission was notable, and should be commented on in the text if a section is not added.  Further, there was nothing on cholangiocarcinoma/gallbladder cancer.  Again worth mentioning, but this omission was not as glaring as that of esophageal adenocarcinoma.

Will need editing for english grammar, especially around verbs; the section on HCC was perhaps the one needing the most attention, but there are grammatical errors throughout.  Please pay special attention to the use of "GC" as an abbreviation for gastric cancer as there were at least two instances of referring to normal GC tissue, which I took to mean non malignant, but GC is by definition malignant as the C stands for cancer.  Spelling throughout good except "squamous" is misspelled in the subtitle for Esophageal Squamous Cell Cancer

Author Response

Point to point reply to the reviewer 3

Very well organized and comprehensive review of topic

My one major critique is the omission of a section on esophageal/GEJ adenocarcinoma - this felt skipped over as there was a section on Gastric Cancer and a section on esophageal squamous cell cancer.  Perhaps there is nothing to review for this type of cancer, but its omission was notable, and should be commented on in the text if a section is not added.  Further, there was nothing on cholangiocarcinoma/gallbladder cancer.  Again, worth mentioning, but this omission was not as glaring as that of esophageal adenocarcinoma.

Will need editing for english grammar, especially around verbs; the section on HCC was perhaps the one needing the most attention, but there are grammatical errors throughout.  Please pay special attention to the use of "GC" as an abbreviation for gastric cancer as there were at least two instances of referring to normal GC tissue, which I took to mean non-malignant, but GC is by definition malignant as the C stands for cancer.  Spelling throughout good except "squamous" is misspelled in the subtitle for Esophageal Squamous Cell Cancer

 Reply: We apologise for being unclear in our manuscript and made a mistake in the subtitle. In fact, Section 3.5 included a paper on esophageal adenocarcinoma, so the subtitle has been changed to “Esophageal Cancer”.  We thank Reviewer#3’s comments and have proofread the manuscript to correct the grammatical mistakes.

Reviewer 4 Report

The authors have set out to review recent literature on the high thoughput proteomics that has been published in GI malignancies.

There is a good summation of the literature.

After reading, several questions/critiques did arise:

  1. When looking at survival, the survival data that is being quoted is out of date. There is more recent analysis of global stats that can be found and should be cited in the manuscript (as 2018 is out of date). Example, SEER database survival for CRC is 14% not less than 10%.
  2. There is mention of the use of NGS, gene expression arrays and high throughput mass spec as technologies used currently. It would be good to compare and contrast the methods so the reader understands the differences between the techniques. why would Mass spec be better or worse than the other technology.
  3. on the first page, you mention therapeutic biomarkers are proteins that can be used for treatment. But therapeutic biomarkers  should also include DNA and RNA based biomarkers, should it not?
  4. You list a number of proteins identified in the proteomic studies but you do not correlate their role in cancer. Do they play a role in angiogenesis? Extravasation? Cell survival? this would be important to add as just having a list of proteins does not really expand the literature in an effective way.
  5. There is mention that in gastric cancer, the use of mass spec could help distinguish stages of gastric cancer (page 6). Why would this be superior or better than using the standard imaging to identify staging?
  6. On page 8, there is mention of an urgent need to establish a non-invasive diagnosis for HCC. But currently, for cirrhotic patients, a triphasic CT of the liver with an elevated AFP is diagnostic for HCC and this is not invasive. The authors should look at the current diagnosis guidelines for the individual GI cancers (such as ASCO, NCCN, ESMO) so that they can compare how proteomic could enhance, improve or be better than standards currently.
  7. I think some time needs to be spent on why is Mass Spec better than a tissue diagnosis.  What are the risks and benefits of both. Can they be additive? As well there is no discussion on the costs to the system (technology costs, bioinformaticians, reporting costs) compared to what is being done now. Would mass spec analysis be quicker than a tissue diagnosis or NGS sequencing?
  8. Would this be a better method of identifying recurrence or diagnosis than what is being working on currently  like circulating tumor cells? I think a little more work on the discussion and future should be spent.

Author Response

Point to point reply to the reviewer 4

The authors have set out to review recent literature on the high throughput proteomics that has been published in GI malignancies.

There is a good summation of the literature.

After reading, several questions/critiques did arise:

1. When looking at survival, the survival data that is being quoted is out of date. There is more recent analysis of global stats that can be found and should be cited in the manuscript (as 2018 is out of date). Example, SEER database survival for CRC is 14% not less than 10%.

Reply: We really thankful to Reviewer #4’s comments. We have updated the recent survival rate according to American Cancer Society in the revised version (Section 3.1 Line 196).

2. There is mention of the use of NGS, gene expression arrays and high throughput mass spec as technologies used currently. It would be good to compare and contrast the methods so the reader understands the differences between the techniques. why would Mass spec be better or worse than the other technology.

Reply: Taking Reviewer #4’s suggestions, we have included a comprehensive comparison emphasizing the advantages of MS over other techniques in the revised manuscript Section 1, Paragraph 4.

3. On the first page, you mention therapeutic biomarkers are proteins that can be used for treatment. But therapeutic biomarkers should also include DNA and RNA based biomarkers, should it not?

Reply: We apologise for the mistake and have revised our statement accordingly (Section 1 Paragraph 1 Line 40). 

4. You list a number of proteins identified in the proteomic studies, but you do not correlate their role in cancer. Do they play a role in angiogenesis? Extravasation? Cell survival? this would be important to add as just having a list of proteins does not really expand the literature in an effective way.

Reply: We understand Reviewer#4’s concern but the review is already too long, and readers may refer to the references for details.  We will consider to arrange the proteins according to their roles in various pathways of carcinogenesis in another manuscript.

5. There is mention that in gastric cancer, the use of mass spec could help distinguish stages of gastric cancer (page 6). Why would this be superior or better than using the standard imaging to identify staging?

Reply: Indeed, some recent publication showed that proteomics-based biomarkers for GC have higher specificity and sensitivity. Therefore, it may be used in clinics if satisfactory results are shown in large cohort of patients.  

6. On page 8, there is mention of an urgent need to establish a non-invasive diagnosis for HCC. But currently, for cirrhotic patients, a triphasic CT of the liver with an elevated AFP is diagnostic for HCC and this is not invasive. The authors should look at the current diagnosis guidelines for the individual GI cancers (such as ASCO, NCCN, ESMO) so that they can compare how proteomic could enhance, improve or be better than standards currently.

Reply: We agree with Reviewer#4 and have deleted the statement mentioning the urgent need of non-invasive tests for HCC, we also agree that AFP used to be a widely accepted biomarker for HCC. However, the reduced specificity and sensitivity of AFP has been reported and its role in the current European and American surveillance guidelines has dropped. Approximately 80% of small HCC (smaller than 3 cm) patients do not show higher AFP level and the sensitivity of using AFP is <25% in this cohort of patients [1]. In addition, AFP negative HCC is very difficult to be distinguished from the AFP-positive HCC [2].

  1. Schütte, K., et al., Current biomarkers for hepatocellular carcinoma: Surveillance, diagnosis and prediction of prognosis. World Journal of Hepatology, 2015. 7(2): p. 139-149.
  2. Wang, T. and K.-H. Zhang, New Blood Biomarkers for the Diagnosis of AFP-Negative Hepatocellular Carcinoma. Frontiers in Oncology, 2020. 10: p. 1316-1316.

7. I think some time needs to be spent on why is Mass Spec better than a tissue diagnosis.  What are the risks and benefits of both. Can they be additive? As well there is no discussion on the costs to the system (technology costs, bioinformaticians, reporting costs) compared to what is being done now. Would mass spec analysis be quicker than a tissue diagnosis or NGS sequencing?

Reply: We should emphasize that we are not proposing to use proteomic analysis or biomarkers to replace any current gold standard for cancer diagnosis.  For sure, the proteomic data will add information to tissue analysis and big scale studies are warranted to address these issues.   We have revised the entire manuscript and tables to address all the issues in the revised version of the manuscript.

8. Would this be a better method of identifying recurrence or diagnosis than what is being working on currently like circulating tumor cells? I think a little more work on the discussion and future should be spent.

Reply: We have updated the discussion part accordingly.

Author Response

Point to point reply to the reviewer 5

Comment-1: Numerous grammatical and syntax errors are present throughout the manuscript and need to be corrected.

Reply: We apologise for the grammatical mistakes and syntax errors, and we have proof-read the manuscript again.

Comment-2: To be more informative, the authors could consider providing clinical performance specifications of each biomarker such as sensitivity, specificity and accuracy based on the information included in the references. It is understandable that most candidate biomarkers have not yet been completely characterized; however, the manuscript would be strengthened by the inclusion of these clinical performance specifications, in the circumstances  where the data are available.

Reply: We really thankful to Reviewer #5’s for his suggestions. We have reviewed and updated the tables including specificity and sensitivity data of the biomarker candidates.

Comment-3: Section 2. MS workflow and recent advancement- the authors mention outdated and infrequently used methods such as 2D-PAGE, SELDI-TOF, and ICAT. The authors should consider adding text addressing ion mobility MS, Single cell proteomics, targeted mass spectrometry, and data-dependent acquisition strategies.

Reply: We appreciate Reviewer #5’s comment. We have updated the recent advances of MS in Section 2, Paragraph 3.

Comment-4: Figure 2. Given that the authors discuss top-down and bottom-up proteomics methods in section 2, the authors should specifically indicate that Figure-2 depicts a typical workflow for bottom-up methods. Additionally, “chemical digestion” should be replaced by “enzymatic digestion”.

Reply: We have revised the title and legend of Figure 2 according to the reviewer’s comments.

Comment-5: Lines 99-101: 2D-DIGE followed by MS analysis is not a top-down approach given that the protein identifications are based on identified peptides.

 Reply: We apologise for the mistake. We have revised the statement and updated the manuscript Section 2, Paragraph 1.

Comment-6: Section 3.2, 2nd paragraph- Please check the accuracy of the units used to describe 2873, 6121, and 7778 vs. 2953 and 1945. Are the correct units m/z or Da?

Reply: Reviewer#5 is correct. The units are in both m/z ratio and Da. We have revised the manuscript accordingly.  

Comment-7: Reference #9 and $10 are identical.

Reply: The duplicated reference has been removed.

Comment-8: Table 1, last row- Q Exactive HF is the name of an instrument and not the name of an MS method.

Reply: We are sorry for the mistake. The method Liquid chromatography-mass spectrometry (HPLC-MS/MS) has been mentioned in the revised version of the manuscript.

Comment-9: Table-3- Provide a deficition of the acronym EThcD-MS/MS

Reply: The abbreviation for EThcD (Electron-transfer/Higher-energy collision dissociation) has been provided in the revised manuscript.

Comment-10: Line 297- TQMS is not a commonly used acronym for tripe quadrupole mass spectrometry. QQQ or QqQ is the preferred acronym.

Reply: The acronym QqQ has been used instead of TQMS in the revised version of the manuscript.

Comment-11: Line 344: spell out ESAD.

Reply: It was supposed to be Esophageal adenocarcinoma and it has been corrected in the revised version.

Round 2

Reviewer 1 Report

Although the manuscript has been partially revised, the overall structure of present article still needs to be improved.

  1. The biomarkers for different GICs could be divided into cells, animals and patients.
  2. The GICs in “Proteomics-based Biomarkers for GICs” part could be arranged according to the order of digestive tract or incidence rate.
  3. Little new information was shown in Figure 1 and Figure 2. It is better to show in supplementary figures.

Author Response

Point to point reply:

1. The biomarkers for different GICs could be divided into cells, animals and patients.

Reply: We thanks to reviewer #1’s suggestions. We have revised the manuscript accordingly.

2. The GICs in “Proteomics-based Biomarkers for GICs” part could be arranged according to the order of digestive tract or incidence rate.

Reply: We thanks to reviewer #1’s comments. The “Proteomics-based Biomarkers for GICs” have already been arranged according to the incidence rate in our manuscript.

3. Little new information was shown in Figure 1 and Figure 2. It is better to show in supplementary figures.

Reply: Both Figure 1 and Figure 2 served as summaries of existing knowledge. Therefore, we prefer to retain them in the main text.

Reviewer 2 Report

All necessary corrections were done in the revised version.

Author Response

Point to point reply:

Comment: All necessary corrections were done in the revised version.

Reply: We really thanks to reviewer #2’s thoughtful review and accepting our revised version for the publication.

Reviewer 4 Report

The authors have responded to my comments and have made the appropriate changes. I have no further questions at this time

Author Response

Point to point reply:

Comments: The authors have responded to my comments and have made the appropriate changes. I have no further questions at this time

Reply: We thanks to reviewer #4’s thoughtful and thorough review of our manuscript and accepting our revision for the publication.

Reviewer 5 Report

  • Lines 118-121: The description of 2D-DIGE is inaccurate. Since the proteins need to be digested following gel electrophoresis and prior to mass spectrometry analysis, this is not considered a top-down method.
  • The use of 2D-DIGE vs. 2D-GE should be consistent throughout the manuscript.
  • Lines 374 & 375: Correct the spelling of "esophargeal"
  • Table 1: MRM needs to be defined

Author Response

Point to point reply:

  • Lines 118-121: The description of 2D-DIGE is inaccurate. Since the proteins need to be digested following gel electrophoresis and prior to mass spectrometry analysis, this is not considered a top-down method.

Reply: We apologise for being unclear and have corrected the statement in the revised manuscript accordingly.

  • The use of 2D-DIGE vs. 2D-GE should be consistent throughout the manuscript.

Reply: We appreciate reviewer #5’s suggestion and we kept “2D-DIGE” consistent throughout the manuscript.

  • Lines 374 & 375: Correct the spelling of "esophargeal"

Reply: We apologise for the mistake and have revised the term accordingly.

  • Table 1: MRM needs to be defined

Reply: We have now defined the MRM in the revised version.